# Evaluation of the Rapid Urease Test (RUT) Device for Rapid Diagnosis of *Helicobacter pylori* in Middle-Aged and Elderly Taiwanese Patients

**DOI:** 10.3390/microorganisms13040767

**Published:** 2025-03-28

**Authors:** Kuan-Yi Yu, Yu-Chuan Chuang, Tien-Yu Huang, Hua-Kang Chou, Ying-Tsang Lu, Juin-Hong Cherng, Cheng-Che Liu

**Affiliations:** 1Graduate Institute of Life Sciences, National Defense Medical Center, Taipei 114, Taiwan; simon791222@gmail.com; 2Division of Gastroenterology, Tri-Service General Hospital Songshan Branch, Taipei 114, Taiwan, rabin.janna@msa.hinet.net (H.-K.C.); 3Division of Gastroenterology, Tri-Service General Hospital, National Defense Medical Center, Taipei 114, Taiwan; tienyu27@gmail.com; 4Pharmacist Association of Taiwan, Taipei 106, Taiwan; andy.lu@strongbiotech.com; 5Department of Biomedical Engineering, Chung Yuan Christian University, Taoyuan 320, Taiwan; 6Department of Physiology and Biophysics, Graduate Institute of Physiology, National Defense Medical Center, Taipei 114, Taiwan

**Keywords:** *Helicobacter pylori*, dry detection device, gel detection device, rapid urease test

## Abstract

The rapid urease test (RUT) is a reliable method for diagnosing *Helicobacter pylori* infections in endoscopy suites; however, there is a need for tests with enhanced sensitivity and faster results. This study aimed to evaluate the diagnostic performance of the new dry detection device test compared to the gel detection device and Pronto Dry RUT in detecting *Helicobacter pylori* infection among middle-aged and elderly Taiwanese individuals. A total of 100 participants with suspected *Helicobacter pylori* infection undergoing upper gastroscopy were prospectively enrolled. The dry detection device demonstrated a 99% concordance rate with the Pronto Dry RUT, with seven participants testing positive for *Helicobacter pylori* using both tests. In contrast, the gel detection device detected only six positive cases, highlighting the superior diagnostic sensitivity of the dry detection device. Additionally, the dry detection device produced significantly faster results than the gel detection device. These findings suggest that the dry detection device is a suitable and efficient RUT for diagnosing *Helicobacter pylori* in middle-aged and elderly patients. Further studies are warranted to explore its application in broader populations and clinical settings.

## 1. Introduction

Although the global incidence of *Helicobacter pylori* infection is declining, it remains highly prevalent worldwide [1]. *Helicobacter pylori* infection is affected by several factors, including age [2]. Some studies have reported that in Taiwan, the prevalence of *Helicobacter pylori* infection is 21.5% in 15–18 year-olds [3], 21.1% in teenagers [4], 54.4% in people over 30 [5], and 59.5% in elderly individuals aged ≥65 [2]. Other studies revealed that the prevalence of *Helicobacter pylori* infection increases with age in both men (20–29-year-olds, 13.7%; 30–39-year-olds, 19.0%; 40–49-year-olds, 26.1%; 50–64-year-olds, 46.2%) and women (20–29-year-olds, 12.7%; 30–39-year-olds, 16.9%; 40–49-year-olds, 26.8%; 50–64-year-olds, 44.1%) [6].

*Helicobacter pylori* is reported as a significant cause of gastrointestinal disorders, including gastritis, peptic ulcers, and gastric cancer [1,7]. Evidence from meta-analyses have demonstrated that *Helicobacter pylori* infection raises the chance of developing hepatocellular carcinoma (HCC) by more than 16 times, cholangiocarcinoma by almost 9 times, and myocardial infarction (MI) by about 2 times [8,9,10]. Animal studies indicated that 37% of Mongolian gerbils developed stomach cancer one year after acquiring *Helicobacter pylori* [11] and that early eradication of *Helicobacter pylori* can minimize the incidence of gastric cancer in hyper gastrinemic mice with *H. pylori* infection [12]. Other available data also show that *Helicobacter pylori* eradication significantly lowers the risk of developing gastric cancer following the endoscopic removal of early-stage gastric cancer compared to placebo treatments [13,14,15]. These findings underscore the critical need for accurate detection of *Helicobacter pylori* infection to ensure effective management and treatment and to prevent the occurrence of further gastrointestinal disorders [16].

The rapid urease test (RUT) is a prominent invasive diagnostic method for the identification of *Helicobacter pylori* that employs stomach mucosal tissue samples placed in a commercially available analysis kit [17], which is aided by its presence in the mucus layer that covers the gastric mucosa [18]. As a non-invasive bacterium, *Helicobacter pylori* lives inside the mucus layer rather than invading the mucosal or submucosal tissues [19]. As a result, the mucus itself is the key component of the RUT, with urease, an enzyme produced by *Helicobacter pylori*, acting as the detection target [20]. Thus, results of the RUT are determined by a color change that occurs within minutes to hours and indicates urease activity [21]. Accurate results require the acquisition of tissue from areas with a large bacterial population and adequate *Helicobacter pylori* levels [22]. However, the use of antibiotics, bismuth-containing medicines, and proton pump inhibitors (PPIs), which reduce bacterial density and induce false-negative results, affects RUT reliability [23]. Furthermore, diseases such as gastric atrophy, intestinal metaplasia, and peptic ulcer bleeding may impair the accuracy of biopsy-based diagnoses [24,25,26]. While multiple biopsies from affected areas have been advocated to improve sensitivity, this technique increases the risk of mucosal damage and associated complications such as bleeding [27].

The gel detection device is an ultrafast RUT that provides results within 5 min of biopsy collection [28]. This rapid turnaround offers a distinct advantage by enabling earlier initiation of treatment. Previous studies have demonstrated the clinical efficacy of gel detection devices in augmenting the endoscopic diagnosis of *Helicobacter pylori* [29]. Meanwhile, the Pronto Dry RUT works by using urea-containing dry filter paper and an indicator that detects a rise in pH in *Helicobacter pylori*-positive samples. The primary benefit of using the Pronto Dry RUT versus a gel detection device in the detection of *Helicobacter pylori* is that the Pronto Dry RUT may be maintained at room temperature, eliminating the requirement to incubate the applied specimen in a warm environment, resulting in faster results [30]. However, limited data are available on the diagnostic performance of the dry detection device, particularly more advanced RUT data on middle-aged and elderly patients. Therefore, this study aimed to prospectively evaluate the sensitivity and specificity of the dry detection device in comparison with the Pronto Dry RUT and gel detection device in symptomatic middle-aged and elderly patients.

## 2. Materials and Methods

### 2.1. Ethical Considerations

This study was approved by the Tri-Service General Hospital Institutional Review Board (approval number: C202205100, 25 November 2022), and all procedures were conducted in accordance with the ethical standards outlined in the Declaration of Helsinki and Good Clinical Practice guidelines. Written informed consent was obtained from all participants prior to their inclusion. Patient data were handled confidentially and used exclusively for research purposes.

### 2.2. Study Population

Based on a high prevalence of *Helicobacter pylori* infection in adults, a total of 100 individuals (both male and female) aged > 18 years with unexplored dyspepsia, characterized by symptoms such as postprandial fullness, early satiety, epigastric pain, and epigastric burning, were enrolled for routine upper gastrointestinal endoscopy in this study. Of the 100 patients, 11% were under 40 years old, 24% were between 40 and 49 years old, 26% were between 50 and 59 years old, 28% were between 60 and 69 years old, 9% were between 70 and 79 years old, and 2% were between 80 and 89 years old. Of these participants, 89 (89%) were middle-aged or elderly, while the remaining 11 (11%) were under 40 years old.

Participants had to be able to provide informed consent and be willing to participate in the study. Exclusion criteria included previous use of antibiotics, proton pump inhibitors, nonsteroidal anti-inflammatory medications, or bismuth salts within three to four weeks of enrollment, as well as a history of gastrointestinal bleeding during the same time period. Patients whose biological samples could not be acquired or who submitted poor-quality samples were also removed. All eligible participants received identical diagnostic tests and reference standards.

### 2.3. Gastroscopy and Sample Collection

During upper gastrointestinal endoscopy, tissue samples were collected using biopsy forceps (2.4 mm diameter and 160 cm length). Samples were obtained from two regions: the lesser curvature of the gastric antrum and the greater curvature of the gastric corpus, targeting areas free of atrophy. Each patient provided four tissue samples for analysis. The samples were subsequently tested using the dry detection device and gel detection device, while the Pronto Dry RUT was used for control testing. The data of the dry detection device and gel detection device were compared to those of the Pronto Dry RUT to produce ratio data.

### 2.4. Culture of Helicobacter pylori

Gastric biopsy samples were treated immediately under microaerophilic settings. To stimulate *Helicobacter pylori* growth, samples were incubated in 9 mL of Tryptic Soy Broth (TSB) supplemented with 5% fetal bovine serum (FBS) at 37 °C for 2 h using the MGC AnaeroPack^®®^-MicroAero system (MITSUBISHI, Tokyo, Japan). Cultures were streaked onto blood agar plates (BAPs) with 5% sheep blood using a three-zone streaking method and cultured at 37 °C for 3–4 days. A single colony was then transferred to 5 mL of TSB for liquid culture under identical conditions. Bacterial growth was measured using optical density at 550 nm (OD550), where OD550 = 0.1 represents 10⁵–10⁶ CFU/mL. The presence of bacteria was confirmed by transmission electron microscopy (TEM) and PCR analysis targeting the *Helicobacter pylori* 16S rRNA gene.

### 2.5. Helicobacter pylori Detection and Confirmation by PCR

To confirm the 16S rRNA of *Helicobacter pylori*, the genomic DNA was extracted from gastric biopsy samples using a TOOLS Fecal DNA Extraction Kit (Cat No. TX-STD01, BIOTOOLS, Taipei, Taiwan). DNA quality and concentration were assessed using a Qubit fluorometer (Invitrogen, Carlsbad, CA, USA). PCR detection of the *Helicobacter pylori* 16S rRNA gene was performed using primers described by Fox et al. [31]: forward primer C97 (5′-GCTATGACGGGTATCC-3′) and reverse primer C98 (5′-GATTTTACCCCTACACCA-3′). The expected amplicon size was 398 bp, and PCR results were verified using 2% agarose gel electrophoresis.

### 2.6. Negative Staining Method for TEM Analysis of Helicobacter pylori

To prepare samples for transmission electron microscopy (TEM), 2% (*w*/*v*), phosphotungstic acid solutions were prepared at pH 6.5 and pH 7.0 as negative staining agents. A drawn capillary tube was used to apply *Helicobacter pylori* onto a carbon-coated copper grid (droplet diameter of approximately 1.5 mm ± 0.5 mm). Excess sample was removed using filter paper, leaving a thin aqueous film. The staining solution was applied for 1 min, and excess liquid was blotted off with filter paper to minimize the residue. Once dried, the samples were observed using TEM under the following conditions: an accelerating voltage of 75 kV and a magnification of 25.0 k.

### 2.7. Statistical Analysis

Sample size calculations were based on an estimated 1 h positivity rate of 70% for the gel detection device and 85% for the dry detection device, assuming an overall *Helicobacter pylori* positivity rate of 80%, a marginal error of 0.05, and a statistical power of 80% [32,33]. Statistical analyses were conducted using chi-square and Fisher’s exact tests. Descriptive statistics were used to present data, expressed as mean ± SD. Sensitivity, specificity, positive predictive value (PPV), and negative predictive value (NPV) were calculated with the requirement of a least two out of three to yield a positive result.

## 3. Results

In this prospective analysis, 100 participants were enrolled, with a mean age of 54.4 ± 12.97 years; 39% of the participants were male. The age distribution was predominantly in the 60–69-year group (28%), followed by 50–59 years (26%) and 40–49 years (24%). Figure 1A illustrates the gastric biopsy sampling process used in this study. Biopsy specimens were collected using endoscopy from two specific stomach regions: zone 1 (lesser curvature) and zone 2 (greater curvature), ensuring comprehensive and representative sampling. Gastric biopsy samples from two randomly selected individuals were subjected to culture, PCR confirmation, and TEM observation, which collectively validated the accuracy of the RUT for detecting *Helicobacter pylori*. Diagnostic testing revealed *Helicobacter pylori* infection in 8 (8%) of the 100 participants, while 92 participants tested negative. Notably, neither the gel detection device nor the dry detection device produced false positives, resulting in a specificity of 100% for both methods. Figure 1B outlines the testing procedure for the dry detection device. Biopsy samples were placed on a test card, treated with the reagent, and securely sealed. Results were assessed by monitoring color changes in the reaction zone, which demonstrated the efficiency and precision of the method. These findings underscore the high reliability of these rapid urease tests for the accurate detection of *Helicobacter pylori* and affirm their clinical utility in ensuring precise diagnosis, which is critical for appropriate treatment and patient care. Additionally, the high specificity minimizes overtreatment, reserving therapeutic measures exclusively for confirmed cases of infection.

Molecular and culture-based analyses provided foundational evidence for *Helicobacter pylori* identification, laying the groundwork for morphological confirmation. PCR targeting the *Helicobacter pylori* 16S rRNA gene successfully amplified a distinct 398 bp amplicon from gastric biopsy samples GAST-S001-HPC and GAST-S002-HPC, as shown in Figure 2A [31] [20]. The negative control (lane 2) showed no amplification, validating the specificity of the primers and the absence of contamination. The expected 400 bp band on the molecular weight marker was highlighted, demonstrating consistency with the reported amplicon size. These findings confirmed the presence of *Helicobacter pylori* in the biopsy isolates.

Furthermore, *Helicobacter pylori* colonies were successfully cultured on blood agar plates (BAPs) under microaerophilic conditions from biopsy samples GAST-S001-HPC and GAST-S002-HPC. Top-down views of the plates showed visible bacterial growth, with colonies appearing small, translucent, and slightly raised (Figure 2B). Close-up images under varying lighting conditions revealed additional structural details of the colonies, marked with yellow arrows for clarity. These findings validated the culture methods employed in this study and confirmed the successful isolation and identification of *Helicobacter pylori*.

Transmission electron microscopy (TEM) provided detailed morphological validation of *Helicobacter pylori* identity. Negative staining consistently revealed the helical shape and polar flagella of the bacteria across multiple fields, confirming the purity and accuracy of the cultured strains. These observations align with previously established descriptions of *Helicobacter pylori* (Figure 3) and provide robust confirmation of its identification.

To evaluate the diagnostic effectiveness of the dry detection device and the gel detection device, Receiver Operating Characteristic (ROC) curve analysis was performed [34] [21]. This statistical tool evaluates the area under the curve (AUC) for urease activity over a 60 min period, facilitating the distinction between *Helicobacter pylori* infection statuses. As shown in Figure 4, the dry detection device achieved an AUC of 0.97, significantly surpassing the 0.88 AUC of the gel detection device. This marked difference highlights the superior diagnostic performance of the dry detection device.

Further analysis evaluated the sensitivity of RUTs at multiple time points after biopsy collection. Figure 5 illustrates the number of positive diagnoses and corresponding sensitivities at various intervals. Notably, there was a progressive increase in positive results as the urease reaction time increased, corroborating findings from previous studies [26]. This pattern underscores the reliability of urease activity as an indicator of the presence of *Helicobacter pylori*, with sensitivity improving over extended reaction times. The incremental increase in positive results reflects the dynamics of urease production by *Helicobacter pylori*, affirming the ability of the test to detect even low levels of bacterial activity when sufficient time is provided.

A detailed comparison of the dry detection device and gel detection device revealed significant advantages of the newer model. As illustrated in Figure 5b, the dry detection device consistently provided faster results and demonstrated greater sensitivity at each evaluated time point compared to its predecessor. These improvements likely stem from advancements in biochemical formulation and processing technology integrated into the dry detection device, which enhance its ability to detect urease, a critical enzyme indicative of the presence of *Helicobacter pylori*. By optimizing detection sensitivity, the dry detection device increases the likelihood of identifying *Helicobacter pylori* at earlier stages with higher accuracy.

The characteristics of enrolled patients, including age, gender, major symptoms, and *Helicobacter pylori* infection rates, as well as endoscopic findings, are presented in Table 1. These data were used as the basis for patient assessment and selection in this study. The Dry detection device exhibited strong performance metrics, including an overall sensitivity of 87.5%, a specificity of 100%, a positive predictive value (PPV) of 100%, and a negative predictive value (NPV) of 98.9%. These findings are summarized in Table 2, which provides comprehensive accuracy data. Notably, the dry detection device achieved optimal performance as early as 10 min post testing, demonstrating the highest sensitivity, with no false positives observed thereafter.

Moreover, the dry detection device consistently demonstrated higher concordance rates with the Pronto dry test at all evaluated time points. This strong agreement between the dry detection device and the Pronto dry test confirms the reliability and efficacy of the former in accurately detecting *Helicobacter pylori* infections.

Overall, these results emphasize the superior performance of the dry detection device in terms of sensitivity, specificity, and concordance with established diagnostic methods. These findings support the dry detection device as an effective tool for the rapid and accurate detection of *Helicobacter pylori*, providing clinicians with a reliable diagnostic option to guide appropriate treatment decisions.

## 4. Discussion

In comparison to the conventional dry rapid urease test (Pronto Dry RUT), we improved the cartridge flow channel structure of the *Helicobacter pylori* detection device to effectively shorten the liquid flow distance, thereby enhancing diffusion efficiency. Subsequently, two new designs for rapid urease tests (RUTs) were developed, aiming to optimize the diffusion rate of the *Helicobacter pylori* urease (HPU) reaction within the device’s containment chamber and to promote uniform reactions between the reagents and gastric biopsy tissues. The dry detection device leverages capillary action caused by perforations in the bottom membrane of the containment chamber, increasing the absorption and diffusion rates of HPU within the chamber. Meanwhile, the gel detection device utilizes a siphon effect driven by the ionic concentration gradient-induced pressure differential between HPU and the reagents. Compared to the dry detection device, this mechanism enables faster diffusion of HPU and its interaction with reagents, significantly reducing the detection time.

In this study, we assessed the diagnostic capabilities of the dry detection device in contrast to its predecessor, the del detection device, and standard quick urease tests like Pronto Dry [35,36]. The dry detection system revealed outstanding diagnostic parameters, including an overall sensitivity of 87.5% and specificity of 100%, with both positive predictive value (PPV) and negative predictive value (NPV) meeting these high expectations [15,37]. These findings support the dry detection device’s promise as a viable diagnostic tool for identifying *Helicobacter pylori* in clinical settings. The dry detection device’s higher sensitivity of 87.5% demonstrates its ability to reliably identify *Helicobacter pylori*, which is crucial for efficient medical treatment [38,39]. The test’s quick performance, with optimum sensitivity achieved in under 10 min, highlights its efficiency and potential to increase patient throughput in clinical settings [40]. Furthermore, the dry detection device’s 100% specificity minimizes false-positive findings, reducing unneeded treatments that might cause patient discomfort and raise healthcare expenses [41]. A 100% PPV verifies that all positive test findings are real infections, guaranteeing that every treated patient actually requires intervention. Similarly, an NPV of 98.9% reduces the likelihood of untreated infections, which might lead to serious problems, including stomach ulcers or cancer.

Our findings also demonstrate a high concordance rate between the dry detection device and the Pronto dry test across all assessed time points, indicating that the dry detection device reliably reproduces the results of other established tests in the market [42]. This consistency is critical in clinical practice, where discrepancies between diagnostic tests can lead to confusion and impede effective treatment planning. The dry detection device’s quick and reliable operation has substantial clinical implications [43]. First, its capacity to produce rapid findings allows for the early commencement of suitable medications, which is especially important for controlling *Helicobacter pylori*-associated illnesses. Early therapy dramatically reduces the likelihood of complications such peptic ulcers and stomach cancer, improving patient outcomes and lowering long-term healthcare costs [44]. Second, the dry detection device’s excellent sensitivity and specificity provide physicians confidence in the results, allowing them to make more educated treatment decisions. This dependability is especially useful in areas with a high frequency of *Helicobacter pylori* infections, where robust and quick diagnostic tools can have a significant influence on public health outcomes.

While this study confirms the efficacy of the dry detection device, it has limitations that warrant further investigation. The number of patients in the study, although sufficient to indicate statistical significance, might be expanded in additional studies to cover more diverse patient groups from various geographic and demographic backgrounds [19]. Likewise, long-term follow-up of patients identified and treated with the dry detection equipment would offer information on the test’s influence on clinical outcomes over time [45]. Future research should also explore the integration of the dry detection device into larger diagnostic frameworks. Combining it with other noninvasive tests for *Helicobacter pylori* could lead to the development of comprehensive diagnostic protocols [46,47]. Comparative studies involving newer versions of rapid urease tests and molecular diagnostic techniques would further define the role of the dry detection device within the evolving landscape of gastrointestinal diagnostics.

## 5. Conclusions

In conclusion, the dry detection device represents a significant advancement in the rapid diagnosis of *Helicobacter pylori* infections. Its high sensitivity, specificity, and rapid turnaround time, combined with excellent concordance with established tests, make it a valuable tool in the clinical management of *Helicobacter pylori*-associated diseases. These features not only improve diagnostic accuracy but also enable timely initiation of appropriate treatment, reducing the risk of complications such as gastric ulcers and cancer. Given the substantial global burden of *Helicobacter pylori* infections, particularly in regions with limited access to advanced diagnostic tools, innovations such as the dry detection device have the potential to transform patient care and public health outcomes.

## Figures and Tables

**Figure 1 microorganisms-13-00767-f001:**
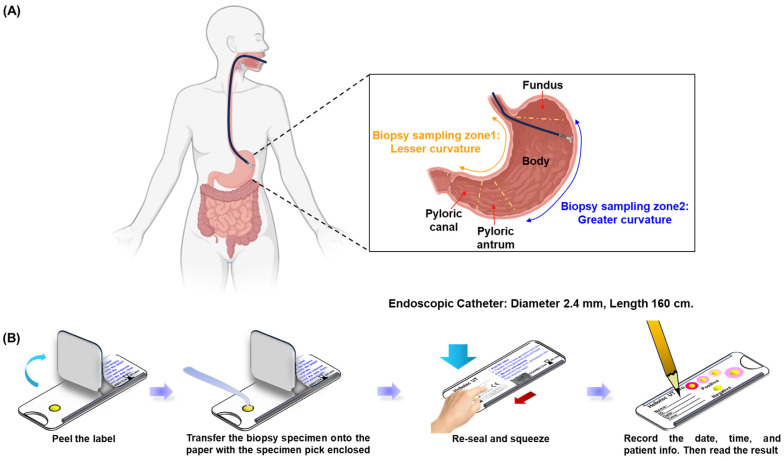
Endoscopic gastric biopsy sampling and diagnostic process. (**A**) Sample collection using endoscopy by scratching the mucus layer of the stomach in zone 1 (lesser curvature) and zone 2 (greater curvature). (**B**) Sample preservation and labeling process.

**Figure 2 microorganisms-13-00767-f002:**
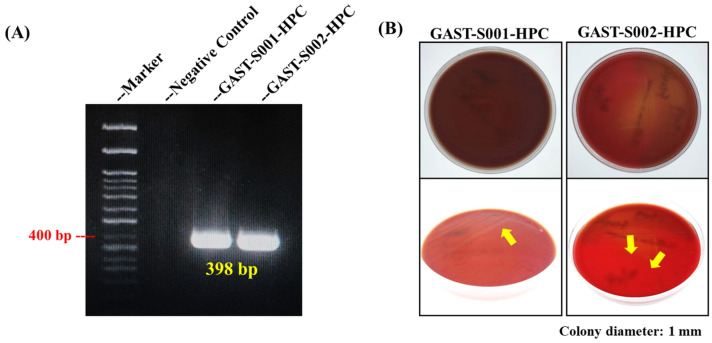
PCR confirmation and culture results of *Helicobacter pylori* from gastric biopsy samples. (**A**) The DNA band in the PCR result confirmed the specific 16S rRNA gene of *Helicobacter pylori.* (**B**) The culture of *Helicobacter pylori* using blood agar. The yellow arrows indicate the locations of *Helicobacter pylori* colonies.

**Figure 3 microorganisms-13-00767-f003:**
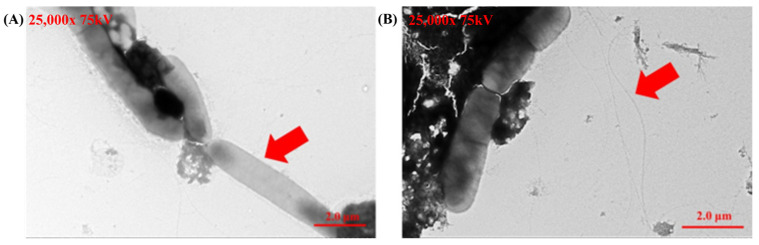
Transmission electron microscopy (TEM) images of *Helicobacter pylori* cultured from gastric biopsy samples. (**A**) The helical shape of *Helicobacter pylori* (showed as the red arrow in **A**). (**B**) The flagella of *Helicobacter pylori* (showed as red arrows in **B**).

**Figure 4 microorganisms-13-00767-f004:**
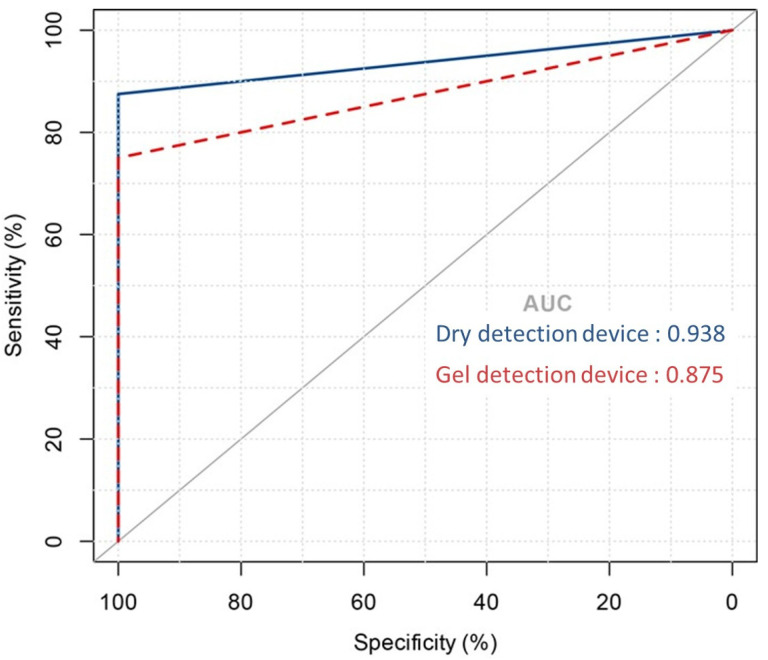
Area under the ROC curve showing the predictive performance of the dry detection device versus the gel detection device model.

**Figure 5 microorganisms-13-00767-f005:**
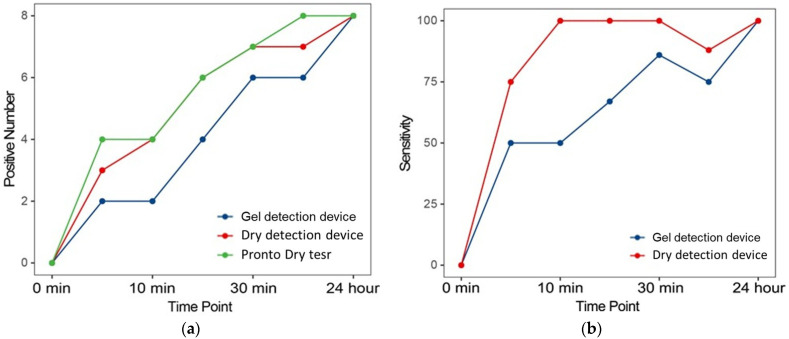
Positive number and sensitivity of each RUT at the different evaluated time points. (**a**) The number of positive diagnoses of lung cancers. (**b**) Corresponding sensitivities at various time points compared to Proto Dry test.

**Table 1 microorganisms-13-00767-t001:** Baseline characteristics of enrolled patients.

Characteristic	Description
Total Number of Participants	100
Gender Distribution	Male: 39%, Female: 61%
Age Range	>18 (Range: 19–89 years)
Mean Age	54.4 ± 12.97 years
Major Symptoms	Unexplained dyspepsia, characterized by postprandial fullness, early satiety, epigastric pain, and epigastric burning.
Endoscopic Sampling Sites	Tissue samples collected from the lesser curvature of the gastric antrum and the greater curvature of the gastric corpus, targeting areas free of atrophy.
*Helicobacter pylori* Infection Rate	8% (8/100 participants diagnosed with *Helicobacter pylori* infection).
Inclusion Criteria	Age 18+~89 years; presence of unexplained dyspepsia; ability to provide informed consent and willingness to participate; undergoing routine upper gastrointestinal endoscopy.
Exclusion Criteria	Prior use of antimicrobials, proton pump inhibitors (PPIs), non-steroidal anti-inflammatory drugs (NSAIDs), or bismuth salts within three to four weeks before enrollment; a history of gastrointestinal bleeding during the same period; inability to obtain biological samples or provision of low-quality samples.

**Table 2 microorganisms-13-00767-t002:** Sensitivity, specificity, PPV, and NPV of RUTs *.

	Dry Detection Device	Gel Detection Device
AUC	Data	Data
Sensitivity	87.5%	75%
Specificity	100%	100%
Negative Predictive Value	98.9%	97.9%
Positive Predictive Value	100%	100%
Accuracy	99%	98%
Precision	100%	100%

* PPV: positive predictive value; NPV: negative predictive value; RUTs: rapid urease tests.

## Data Availability

The raw data supporting the conclusions of this article will be made available by the authors on request.

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
