# Peer review of "Evaluation of the Rapid Urease Test (RUT) Device for Rapid Diagnosis of *Helicobacter pylori* in Middle-Aged and Elderly Taiwanese Patients"

_microorganisms, 2025, doi:10.3390/microorganisms13040767_

Round 1
Reviewer 1 Report
Comments and Suggestions for Authors
I reviewed a paper titled: “Evaluation of the Devise of Rapid Urease Test (RUTs) for Rapid Diagnosis of Helicobacter pylori in Middle-Aged and Elderly Taiwanese Patients”. The major concerns about this manuscript are as below;
1) ‘Devise’ in the manuscript title may be misprint and should be corrected.
2) In line 79 of the Method section, the authors described the inclusion criteria to be > 18 years. Please explain the reason why only middle aged and elderly subjects were enrolled into this study.
3) Pronto Dry RUT was missing in lines 61-67 of the Introduction section.
4) When calculating the sample size, the authors described in lines 126-127 as below.
‘….an estimated 1-hour positivity rate of 70% for the Gel detection device and 85% for the Dry detection device, assuming an overall H. pylori positivity rate of 80%...’
Please cite the references about the previous studies.
5) Please create Table 1 for baseline characteristic of enrolled patients, including endoscopic findings.
6) I don’t understand the meaning of ‘bracket’ in line 137.
7) Please clarify the gold standard for H. pylori diagnosis in this study (Pronto Dry RUT or PCR or TEM?).
8) In this study, H. pylori infection rate was very low of 8% (n = 8/100). I think that the number of patients was too small to evaluate the diagnostic performance of H. pylori.
9) iThenticate reported the percent check of 34%. Generally, a similarity of > 30% is considered very high risk of plagiarism.
Author Response
Thank you for your thorough and insightful review of our manuscript, titled "Evaluation of the Rapid Urease Test (RUTs) Device for Rapid Diagnosis of Helicobacter pylori in Middle-Aged and Elderly Taiwanese Patients." We greatly appreciate the time and effort you dedicated to providing constructive feedback, which has been invaluable in improving the quality of our work.
We have carefully considered each of your comments and have made significant revisions to the manuscript accordingly. Below, we address each of your points in detail:
1) ‘Devise’ in the manuscript title may be misprint and should be corrected.
Answer: We've revised 'Devise' to ‘Device’, thank you!
2) In line 79 of the Method section, the authors described the inclusion criteria to be > 18 years. Please explain the reason why only middle aged and elderly subjects were enrolled into this study.
Answer: Thank you for your question. In the section of Introduction, we have added the age range people who suffer from H. pylori at lines 37-44 as supporting previous statements.
Additionally, we actually recruited 100 male and female participants, of which 11% were under 40 years old, 24% were between 40 and 49 years old, 26% were between 50 and 59 years old, 28% were between 60 and 69 years old, 9% were between 70 and 79 years old, and 2% were between 80 and 89 years old. Of these participants, 89 (89%) were middle-aged or elderly, while the remaining 11 (11%) were under 40 years old. Therefore, we recruited not only middle-aged and elderly individuals but also participants from other age groups.
Since the majority of participants were concentrated in the 40–69 age range, all of them had unexplained dyspepsia, characterized by symptoms such as postprandial fullness, early satiety, epigastric pain, and epigastric burning, and underwent routine upper gastrointestinal endoscopy. However, the focus of this study was on middle-aged and elderly participants, as this age group is more prone to Helicobacter pylori infection, which can lead to various gastrointestinal diseases such as dyspepsia, peptic ulcers, and gastric cancer.
Furthermore, the prevalence of Helicobacter pylori infection is generally higher in the elderly population, making them a key group for evaluating the diagnostic performance of the Rapid Urease Test (RUT) in a clinical setting. Despite the decreasing global prevalence of Helicobacter pylori infection, it remains highly prevalent worldwide and is a common cause of gastrointestinal diseases in middle-aged and elderly populations. Therefore, evaluating diagnostic methods for this specific group is of significant clinical importance and reflects the core target population of our study.
3) Pronto Dry RUT was missing in lines 76-83 of the Introduction section.
Answer: Thank you for reminding us. The explanation about Pronto Dry RUT is already added at line 76 to 83.
4) When calculating the sample size, the authors described in lines 126-127 as below.
‘….an estimated 1-hour positivity rate of 70% for the Gel detection device and 85% for the Dry detection device, assuming an overall H. pylori positivity rate of 80%...’
Please cite the references about the previous studies.
Answer: Thank you for your suggestion. We have cited this sentence at lines 154.
5) Please create Table 1 for baseline characteristic of enrolled patients, including endoscopic findings.
Answer: Thank you for your suggestion. We have already added this table at line 260.
Table 1. Baseline Characteristics of Enrolled Patients.
6) I don’t understand the meaning of ‘bracket’ in line 137.
Answer: In this context, "bracket" refers to an age range or age group. It is commonly used to categorize a set of numbers within a specific range. In the sentence, "The age distribution was predominantly in the 60-69-year bracket," it means that the participants in the study were mostly concentrated within the 60 to 69 age range, or the majority of the participants fell within this specific age interval. So, "bracket" here refers to a specific age range or category. To avoid any confusion, we’re pleased to revise it to: 'The age distribution was predominantly in the 60-69-year age group (28%), followed by the 50-59 years (26%) and 40-49 years (24%).
7) Please clarify the gold standard for H. pylori diagnosis in this study (Pronto Dry RUT or PCR or TEM?).
Answer: The gold standard for H. pylori diagnosis in this study was a composite criterion, not a single test.
A positive H. pylori infection was defined as having at least two positive results from the three main Rapid Urease Tests (RUTs):
- Dry detection device (Helicotech UT Pro)
- Gel detection device (Helicotech UT Plus)
- Pronto Dry RUT
To further validate the presence of H. pylori, the study also utilized:
- H. pylori culture
- PCR targeting the 16S rRNA gene at line 160 to 167
- Transmission electron microscopy (TEM) at line 177 to 181
The Pronto Dry RUT, used as a control, demonstrated high concordance with the Dry detection device.
In summary, the study employed a comprehensive approach, relying on multiple RUTs and traditional methods to confirm H. pylori infection and enhance the diagnostic accuracy of the Dry detection device.
8) In this study, H. pylori infection rate was very low of 8% (n = 8/100). I think that the number of patients was too small to evaluate the diagnostic performance of H. pylori.
Answer:
Thank you for your insightful comments regarding the 8% Helicobacter pylori infection rate (n=100) observed in our study. We understand your concern about whether this rate is sufficient for evaluating diagnostic performance.
First, we would like to clarify that the primary objective of this study was to evaluate the diagnostic performance of the new Dry detection device, not to assess the general prevalence of H. pylori. Our study population consisted of middle-aged and elderly Taiwanese patients with unexplained dyspepsia undergoing upper gastrointestinal endoscopy. As a result, the infection rate in this group may differ from that of the general population.
The sample size of 100 participants was determined based on a statistical power analysis conducted prior to the study, which took into account the expected positivity rates of the Dry detection device (85%) and the Gel detection device (70%), as well as the overall H. pylori. positivity rate (80%). Despite the lower infection rate observed, this sample size was sufficient for comparing the diagnostic performance of the two devices.
While the number of positive cases was small, the Dry detection device demonstrated 100% specificity and strong concordance with the Pronto Dry RUT across all time points, supporting its reliability. Accurate diagnosis in this group remains clinically significant, as eradicating H. pylori can reduce the risk of gastric cancer.
We acknowledge that the low infection rate limits of our ability to fully assess sensitivity. In future research, we plan to:
- Increase the sample size to include more H. pylori.-positive individuals.
- Expand studies to diverse geographical and demographic populations.
- Conduct long-term follow-up studies to evaluate clinical outcomes.
In conclusion, despite the lower infection rate, the high specificity and strong concordance with the reference method demonstrate the reliability of the Dry detection device in our population. We recognize the need for larger-scale studies to more comprehensively evaluate its sensitivity.
9) iThenticate reported the percent check of 34%. Generally, a similarity of > 30% is considered very high risk of plagiarism.
Answer: Thank you very much for your warning. Our manuscript is already revised and paraphrased. We hope that the new version is now at low percentage of similarity.
Reviewer 2 Report
Comments and Suggestions for Authors
I had difficulty liking the manuscript because the goal was unclear from the introduction. However, after reading the discussion, I finally understood the goal.
1) In terms of novelty, comparing two rapid diagnostic tests seems a bit mundane.
2) If the goal of the manuscript was to compare the Dry detection test and the Gel detection test, using the Pronto Dry as a reference. Why Table 1 does not include the Pronto dry results?
3) Same for Figure 4. Data on Pronto Dry is missing.
3) Figure 3, the TEM seems a bit unnecessary. It provides very little information.
4) If the goal of a rapid urease test is diagnostic time, the gel detection (Helicotec UT) seems less effective despite all three assays having the same sensitivity after 24 hours.
5) In Figure 5b, the Pronto Dry test data is missing.
6) Minor comments. The name of the test is HelicotecUT PLus and HelicotecUT. Also, check that H. pylori is italicized throughout the manuscript.
7. Alimnoitation of rapid urease test is the requirement for large amounts of bacteria in the specimen. This is difficult to control as you use different patients or collect biopsies from different regions in the stomach. Nevertheless, negative results could mean that the level of bacteria in the specimen obtained is low or that positive results will take longer for the change in color to take place.
Author Response
Thank you for your thorough and insightful review of our manuscript, titled "Evaluation of the Rapid Urease Test (RUTs) Device for Rapid Diagnosis of Helicobacter pylori in Middle-Aged and Elderly Taiwanese Patients." We greatly appreciate the time and effort you dedicated to providing constructive feedback, which has been invaluable in improving the quality of our work.
We have carefully considered each of your comments and have made significant revisions to the manuscript accordingly. Below, we address each of your points in detail:
1) In terms of novelty, comparing two rapid diagnostic tests seems a bit mundane.
Answer: Thank you for your comments regarding the novelty of our study. While comparing two rapid diagnostic tests might seem a bit mundane at first glance, our research incorporates several novel aspects that contribute significantly to the field:
The Dry detection device offers significant advancements over both the Gel Detection Device and the traditional Pronto Dry RUT. One key innovation lies in its cartridge flow channel structure, which is designed to shorten the liquid flow distance and enhance diffusion efficiency. This modification allows for a more effective and uniform interaction between the urease (HPU) and reagents, resulting in faster and more reliable diagnostic outcomes.
In contrast to the Gel Detection Device, which utilizes a siphon effect driven by an ionic concentration gradient to expedite diffusion, the Dry detection device leverages capillary action created by perforations in the containment chamber's membrane. This mechanism significantly increases the absorption and diffusion rates of HPU within the device, providing results with high sensitivity (87.5%) and 100% specificity in as little as 10 minutes. The Dry detection device’s ability to eliminate false positives (100% specificity) and accurately identify true infections (100% PPV) makes it a reliable tool for H. pylori diagnosis, especially in clinical settings where rapid and accurate results are crucial.
When compared to Pronto Dry, the Dry detection device maintains a high concordance rate across all assessed time points, demonstrating that it can reproduce results comparable to established diagnostic tests. This reliability strengthens its position as a competitive option for H. pylori diagnosis in clinical environments.
This study not only compares two rapid diagnostic methods but also systematically evaluates the Dry rapid urease test device based on innovative design principles, particularly in middle-aged and elderly populations in Taiwan. By comparing it with the Gel detection device and traditional methods, we highlight the potential technological advantages and clinical value of the Dry detection device. These findings offer important insights into the rapid diagnosis of Helicobacter pylori infections, providing a valuable tool for clinicians and improving patient care efficiency.
2) If the goal of the manuscript was to compare the Dry detection test and the Gel detection test, using the Pronto Dry as a reference. Why Table 1 does not include the Pronto dry results?
Answer: Thank you for your question. Pronto Dry RUT is a commercial product and used as the control group. The values of Dry detection device and Gel detection device groups were compared with the Pronto Dry Rut in the form of a ratio. Thus, the Pronto Dry RUT values do not need to be displayed.
3) Same for Figure 4. Data on Pronto Dry is missing.
Answer: As we mentioned before, the Pronto Dry RUT is a commercial product and used as the control group. The values of Dry detection device and Gel detection device groups were compared with the Pronto Dry Rut in the form of a ratio. Thus, the Pronto Dry RUT values do not need to be displayed.
3) Figure 3, the TEM seems a bit unnecessary. It provides very little information.
Answer: In this study, TEM figure is necessary to confirm morphological evidence of the isolated H. pylori from gastric biopsy samples.
4) If the goal of a rapid urease test is diagnostic time, the gel detection (Helicotec UT) seems less effective despite all three assays having the same sensitivity after 24 hours.
Answer: The goal of the study is to identify an optimal urease test, not only rapidity, but also to both accuracy and sensitivity, thereby ensuring reliable and early detection of urease. Our findings, derived from a comparative analysis, demonstrate that a Dry detection device outperformed a Gel detection device, exhibiting better sensitivity that providing a more efficient and effective diagnostic tool for clinical applications.
5) In Figure 5b, the Pronto Dry test data is missing.
Answer: As previously stated, the Pronto Dry RUT is a commercial product and serves as a control group. The data of the Dry and Gel detection device groups were compared to the Pronto Dry Rut using a ratio. Thus, the Pronto Dry RUT values are not required to be displayed. The explanation was added at lines 113-114.
6) Minor comments. The name of the test is HelicotecUT PLus and HelicotecUT. Also, check that H. pylori is italicized throughout the manuscript.
Answer: Thank you very much for your suggestion. For consistency, we have standardized the terms "Helicotec UT Plus" and "Helicotec UT" in the article to "Dry Detection" and "Gel Detection," respectively. Furthermore, the article has been revised to consistently use Helicobacter pylori in italics instead of H. pylori, except in the references.
7) Alimnoitation of rapid urease test is the requirement for large amounts of bacteria in the specimen. This is difficult to control as you use different patients or collect biopsies from different regions in the stomach. Nevertheless, negative results could mean that the level of bacteria in the specimen obtained is low or that positive results will take longer for the change in color to take place.
Answer: Thank you for highlighting the limitation of the rapid urease test (RUT) regarding the need for a large amount of bacterial in the specimen. This is indeed challenging to control, particularly due to variations between patients and the specific locations of biopsies within the stomach. As you pointed out, negative results could indicate a low bacterial load in the sample, while positive results might take longer to appear due to a delayed color change, especially if the bacterial density is lower. Additionally, factors like the use of antibiotics, bismuth compounds, and proton-pump inhibitors (PPIs) can reduce bacterial levels, potentially leading to false negatives.
In our study, to address these issues, we employed a multi-site sampling approach, collecting tissue samples from two specific regions in the stomach—the lesser curvature of the gastric antrum and the greater curvature of the gastric corpus—targeting areas free from atrophy. This approach aimed to increase the likelihood of detecting H. pylori, even in cases of uneven bacterial distribution within the stomach.
Our results demonstrated that the new Dry detection device exhibited higher sensitivity and produced faster results compared to the Gel detection device. The Dry detection device achieved optimal performance as early as 10 minutes post-testing, which may partially address the issue of delayed positive results due to lower bacterial loads. Its higher sensitivity (87.5%) also suggests a potentially better ability to detect lower levels of bacteria compared to traditional methods.
Furthermore, while the RUT has inherent limitations, our study also included complementary diagnostic methods, such as bacterial culture, PCR, and transmission electron microscopy (TEM), to validate the RUT results. These complementary methods helped enhance the overall diagnostic accuracy, making the study more robust in addressing the challenges associated with bacterial load and sampling location.
We hope this approach provides insight into how we aimed to optimize rapid diagnosis while acknowledging the limitations of the RUT.
Reviewer 3 Report
Comments and Suggestions for Authors
Review for the manuscript
Journal:
Microorganisms (ISSN 2076-2607)
Manuscript ID:
microorganisms-3453368
Type:
Article title:
Evaluation of the Devise of Rapid Urease Test (RUTs) for Rapid Diagnosis of Helicobacter pylori in Middle-Aged and Elderly Taiwanese Patients
Section:
Medical Microbiology
Special Issue:
Helicobacter pylori Infection: Detection and Novel Treatment
Dear Editor,
Thank you for the invitation to review for Microorganisms. I have some comments and suggestions for this manuscript before it can be considered for publication.
OVERALL COMMENTS
- Based on the statement that the rapid urease test is a reliable method for diagnosing Helicobacter pylori ( pylori) infections in endoscopy suites; however, there is a need for tests with enhanced sensitivity and faster results, the authors of this study aimed to evaluate the diagnostic performance of the new Dry detection device test compared to the Gel detection device and Pronto Dry RUT in detecting H. pylori infection among middle-aged and elderly Taiwanese individuals. A total of 100 participants with suspected H. pylori infection undergoing upper gastroscopy were prospectively enrolled. The Dry detection device demonstrated a 99% concordance rate with the Pronto Dry RUT, with seven participants testing positive for H. pylori using both tests. In contrast, the Gel detection device detected only six positive cases, highlighting the superior diagnostic sensitivity of the Dry detection device. The Dry detection device also produced significantly faster results than the Gel detection device. These findings suggest that the Dry detection device is a suitable and efficient RUT for diagnosing H. pylori in middle-aged and elderly patients. Further studies are warranted to explore its application in broader populations and clinical settings.
- I suggest that the authors review the language. There are some mistakes regarding punctuation, grammar, etc.
TITLE
The title is “Evaluation of the Devise of Rapid Urease Test (RUTs) for Rapid Diagnosis of Helicobacter pylori in Middle-Aged and Elderly Taiwanese Patients”. It is adequate.
However, would it be “device” instead of “devise”?
ABSTRACT
This section is adequate.
KEYWORDS
The keywords are adequate.
INTRODUCTION
Please include newer references in this section. Please visit PUBMED.com, where you can find very good studies published in 2024-2025.
METHODS
This section was well performed.
Did the authors use a sample calculation? Please justify.
RESULTS
Please use italics for Helicobacter pylori in the legend of Figure 2. The same in line 174. Please check along with the entire text. See that the legend of Figures 2 and 3 brings H. pylori and Helicobacter pylori. Please use a standard.
Please expand the legend of Figure 1.
DISCUSSION
This section is adequate. The results of the study are well documented. However, I suggest the inclusion of newer references.
Why can't the devise be used by people of other ages?
CONCLUSIONS
This section is adequate for the study findings. The limitations are also adequate, as well as the future perspectives (lines 273-284).
REFERENCES
It is necessary to include newer references in this section.
Other comments
- Please check that there are some Grammar mistakes along with the text,
- pylori or Helicobacter pylori appears without italics in many parts of the text, mainly in the Discussion (example: lines 33, 164, 174, 177, 184, 192, 194, 217…)
- Sometimes in the text, the authors use H. pylori and sometimes Helicobacter pylori (Please see the legend of Figures 2 and 3 as examples). After the first time H. pylori is used, the same format should be used in the rest of the text. Please check.
I wish the authors good luck with this manuscript.
Comments on the Quality of English LanguageSome little mistakes need to be corrected.
Author Response
Thank you for your thorough and insightful review of our manuscript, titled "Evaluation of the Rapid Urease Test (RUTs) Device for Rapid Diagnosis of Helicobacter pylori in Middle-Aged and Elderly Taiwanese Patients." We greatly appreciate the time and effort you dedicated to providing constructive feedback, which has been invaluable in improving the quality of our work.
We have carefully considered each of your comments and have made significant revisions to the manuscript accordingly. Below, we address each of your points in detail:
1) TITLE
The title is “Evaluation of the Devise of Rapid Urease Test (RUTs) for Rapid Diagnosis of Helicobacter pylori in Middle-Aged and Elderly Taiwanese Patients”. It is adequate.
- However, would it be “device” instead of “devise”?
Answer: Thank you very much for your correction. We revised ‘Devise’ to ‘Device’ accordingly.
2) ABSTRACT
- This section is adequate.
Answer: Thank you for your feedback. We appreciate your confirmation that the abstract section is adequate.
3) KEYWORDS
- The keywords are adequate.
Answer: Thank you for your feedback. We appreciate your confirmation that the keywords are adequate.
4) INTRODUCTION
- Please include newer references in this section. Please visit PUBMED.com, where you can find very good studies published in 2024-2025.
Answer: Thank you very much for your suggestions. We have added several new references in the manuscript. However, some of the old references are important to support the fundamental of our study, so we retain it in our manuscript.
5) METHODS
- This section was well performed. Did the authors use a sample calculation? Please justify.
Answer: Thank you for your attention to our study methodology. Yes, a sample size calculation was clearly performed and detailed in the Statistical Analysis (section 2.7) of the Methods section.
The primary justification for conducting the sample size calculation was to ensure that our study had sufficient statistical power to reliably evaluate the diagnostic performance of the Dry detection device. The calculation was based on the following estimated values and statistical parameters:
- An estimated 1-hour positivity rate of 70% for the Gel detection device.
- An estimated 1-hour positivity rate of 85% for the Dry detection device.
- An assumed overall pylori. positivity rate of 80%.
- A set margin of error of 0.05.
- A desired statistical power of 80%.
Based on these parameters, we calculated that a total of 100 participants were necessary for this study. We believe this sample size provides adequate statistical power to assess the diagnostic efficacy of the Dry detection device compared to the Gel detection device, considering the precision of our findings.
6) RESULTS
- Please use italics for Helicobacter pylori in the legend of Figure 2. The same in line 174. Please check along with the entire text. See that the legend of Figures 2 and 3 brings pylori. and Helicobacter pylori. Please use a standard.
Answer: Thank you for your suggestion. The article has been revised to consistently use Helicobacter pylori in italics instead of H. pylori, except in the references.
7) Please expand the legend of Figure 1.
Answer: Thank you for your suggestion. We have expanded the legend in the Figure 1, 2, and 3 to serve more detail explanation.
DISCUSSION
8) This section is adequate. The results of the study are well documented. However, I suggest the inclusion of newer references. Why can't the devise be used by people of other ages?
Answer: Thank you for your question. As you can see, our inclusion criteria are individuals over the age of 18, our device is suitable for use across other age groups. We recruited a total of 100 participants, with 11% of participants being under 40 years old, 24% between 40 and 49 years old, 26% between 50 and 59 years old, 28% between 60 and 69 years old, 9% between 70 and 79 years old, and 2% between 80 and 89 years old.
The actual age distribution of the recruited participants shows that only 11% were under 40, with the majority of participants concentrated in the 40–69 age range. This reflects the target population of our study. Although the global prevalence of Helicobacter pylori infection is decreasing, it remains highly prevalent worldwide and continues to be a common cause of gastrointestinal disease in middle-aged and elderly populations. Evaluating diagnostic methods for this specific group is of significant clinical importance.
The principle of the Dry detection device is based on detecting the urease activity produced by Helicobacter pylori. In theory, as long as there is Helicobacter pylori infection, patients of any age group can potentially be detected using this method.
9) CONCLUSIONS
This section is adequate for the study findings. The limitations are also adequate, as well as the future perspectives (lines 273-284).
Answer: Thank you for your feedback. We appreciate your positive comments regarding the conclusions and the adequacy of the limitations and future perspectives (lines 273-284). We are glad that these sections meet your expectations.
10) REFERENCES
It is necessary to include newer references in this section.
Answer: Thank you very much for your suggestions. As we mentioned before, we have added several new references in the manuscript. However, some of the old references are important to support the fundamental of our study, so we retain it in our manuscript.
11) OTHER COMMENTS
Please check that there are some Grammar mistakes along with the text, pylori or Helicobacter pylori appears without italics in many parts of the text, mainly in the Discussion (example: lines 33, 164, 174, 177, 184, 192, 194, 217…)
Sometimes in the text, the authors use H. pylori and sometimes Helicobacter pylori (Please see the legend of Figures 2 and 3 as examples). After the first time H. pylori is used, the same format should be used in the rest of the text. Please check. I wish the authors good luck with this manuscript.
Answer: Thank you very much for your suggestions. We have revised the grammar mistakes in our manuscripts, and the article has been revised to consistently use Helicobacter pylori in italics instead of H. pylori, except in the references.
12) Comments on the Quality of English Language
Some little mistakes need to be corrected.
Answer: Thank you for your correction. We have revised the language mistakes.
Round 2
Reviewer 1 Report
Comments and Suggestions for Authors
The authors answered to the comments 1-8.
Reviewer 2 Report
Comments and Suggestions for Authors
The manuscript is much improved, and although novelty is limited, this manuscript can provide valuable information to health and research clinics. You don't need to spell out Helicobacter; just ensure that when using the term H. pylori, it is always italicized.